# Microorganisms and Breast Cancer: An In-Depth Analysis of Clinical Studies

**DOI:** 10.3390/pathogens13010006

**Published:** 2023-12-19

**Authors:** Naghmeh Naderi, Afshin Mosahebi, Norman R. Williams

**Affiliations:** 1Department of Plastic and Reconstructive Surgery, Royal Free Hospital, London NW3 2QG, UK; naghmeh.naderi@nhs.net (N.N.); a.mosahebi@ucl.ac.uk (A.M.); 2Division of Surgery & Interventional Science, University College London, London W1W 7TY, UK

**Keywords:** female, breast neoplasms, fungi, bacteria, viruses

## Abstract

Breast cancer is a multifactorial disease that affects millions of women worldwide. Recent work has shown intriguing connections between microorganisms and breast cancer, which might have implications for prevention and treatment. This article analyzed 117 relevant breast cancer clinical studies listed on ClinicalTrials.gov selected using a bespoke set of 38 search terms focused on bacteria, viruses, and fungi. This was supplemented with 20 studies found from a search of PubMed. The resulting 137 studies were described by their characteristics such as geographic distribution, interventions used, start date and status, etc. The studies were then collated into thematic groups for a descriptive analysis to identify knowledge gaps and emerging trends.

## 1. Introduction

There is a large and diverse population of microorganisms in the human body that can be either harmful to health or remedial [1,2,3]. Consequently, there has been a drive to fully characterize the microorganisms associated with different organs under different health conditions. While there has been some debate about whether microbial differences are a consequence or a cause of the disease, there is evidence to favor the latter [4,5]. The microbiome has been implicated at a variety of body sites, including the skin, gut, pancreas, liver, lung, prostate, and breast [6]. It can modify the normal biological function of human organs and influence the likelihood of the development of non-infectious autoimmune diseases, such as diabetes mellitus and inflammatory bowel disease, as well as various organ-specific cancers [6]. Mechanisms by which microbial environments alter cellular function include influencing immunological function [7], small-signaling-metabolite synthesis [8], energy-harvest efficiency [9], and circulating steroid hormone levels [10]. Certain microbial species can cause DNA damage, mutations, and epigenetic modifications [11]. Breast tissue has been shown to have a distinct microbiome that is different from that of other body sites [8]. Proteobacteria, followed by firmicutes, are the most abundant phylum represented in breast tissue [12]. 

Diseases of the breast can be either benign or malignant with the latter accounting for one-quarter of all cancers worldwide [13]. Breast cancer is likely to be caused by a complex interplay between genetic and environmental factors. There remain many uncharacterized mechanisms that promote its development. It has been well-established that there are microbes in breast cancer tissue, which are distinct from that of benign breast tissue. Hieken et al. showed that breast malignancy correlated with enrichment in taxa of lower abundance, including the genera *Fusobacterium*, *Atopobium*, *Gluconacetobacter*, *Hydrogenophaga* and *Lactobacillus* [4]. Urbaniak et al. showed higher relative abundance of *Bacillus*, *Enterobacteriaceae*, and *Staphylococcus* in breast cancer tissue. They also showed that *Escherichia coli* and *Staphylococcus epidermidis* isolated from breast cancer tissue induced DNA double-stranded breaks in HeLa cells [8]. The role of the microbiome in breast cancer is an area of intensive research with antimicrobials and/or probiotics potentially playing a role in the prevention and treatment of breast disease [14]. There are several clinical trials investigating the role of microorganisms in breast cancer and their potential as components of risk-based breast cancer screening and primary prevention [15]. 

The ClinicalTrials.gov database is free to use and contains details of more than 450,000 privately and publicly funded clinical studies conducted in 221 countries [16] with the number of studies registered increasing exponentially [17]; an analysis of these studies can identify knowledge gaps and emerging trends in research. Not all studies are registered, so it is important to supplement this information with a search of the literature. The PubMed database is a free resource that contains more than 36 million citations and abstracts from the biomedical literature [18].

This article describes the past, ongoing, and planned clinical trials of microorganisms and breast cancer, together with a thematic analysis. This information will be useful to those considering designing or participating in current or future trials.

## 2. Materials and Methods

Specific search terms were used in the query section of ClinicalTrials.gov. A comma-separated values (CSV) file of all studies meeting the query criteria was downloaded and converted to Microsoft^®^ Excel^®^ for Microsoft 365 for analysis by JMP^®^ Pro 17.2.0 running on Windows 10. The titles of every study in this file were then searched for keywords relevant to “microorganisms”, and this list of terms was used to select relevant studies which were then analyzed.

In addition, a literature review in PubMed was performed to capture studies that may not be listed on ClinicalTrials.gov.

## 3. Results

### 3.1. Analysis of Studies Reported in ClinicalTrials.gov and PubMed

#### 3.1.1. Selection of Relevant Terms and Studies

An initial query of “breast cancer” in “condition or disease” and ”microorganism” in “other terms” found 25 studies, but from the authors’ experience it was evident that several studies were not included. Therefore, it was decided to construct bespoke search terminology to widen the scope of studies found, while keeping within the topic.

A query of “breast cancer” in “condition or disease” on 14 July 2023 of 458,710 studies listed on ClinicalTrials.gov found 13,080 studies containing 10,729 terms in the study titles. These terms were individually assessed by the author for their relevance to the topic. After several iterations, 38 terms were selected that identified 122 studies. A further four studies were removed as they contained terms that on closer inspection were not relevant to the topic (“mycosis fungoides”, “Motive Flora”, ”non-viral”). One study used the phrase “v-ERB-B2 Avian Erythroblastic Leukemia Viral Oncogene Homolog 2” to describe HER2, so this was also removed. This yielded 117 studies. Figure 1 is a flow diagram illustrating these steps. Figure 2 shows a word cloud of the search terms with more than one “hit”, and Table 1 lists all the search terms used.

In addition, a literature review using the terms “breast” AND “microbiota” AND (“Case-Control Study” OR “Clinical Trial”) in PubMed yielded 91 results. Of these, 71 were excluded, leaving 20 to be added to the 117 studies found by the ClinicalTrials.gov search. Figure 3 is a flow diagram illustrating these steps.

In total, 137 studies were taken forward for analysis. Details for each of these studies can be found in Appendix A.

#### 3.1.2. Geographic Distribution

The geographic distribution of the studies is shown in Figure 4 and listed in Table 2. Most studies were run in countries where the PI (Principal Investigator) was based in the USA (75/137 = 55%), followed by China (18/137 = 13%).

#### 3.1.3. Status of Studies

The status of the studies is shown in Table 3. Just over half (51% = 70/137) of studies had “completed” (the study had ended normally, and the last participant’s last visit had occurred), and 30% (41/137) were either “recruiting” or “active and not recruiting”. One study (NCT04395508) had been approved for marketing. The eight studies that had been “terminated” had stopped early and will not start again. The six studies marked as “withdrawn” stopped before enrolling the first participant. The eight studies marked as “unknown” had a last known status of “recruiting”, “not yet recruiting”, or “active and not recruiting” but have passed their completion dates, and the status had not been verified within the past two years.

Only 21% (29/137) of the studies had results available (Table 4).

#### 3.1.4. Start Date

The start dates of the studies are shown in Figure 5 and range from 1995 to 2023. Twelve studies did not report a start date. 

Details of 13 studies with a planned start date in 2022 and 2023 are shown in Table 5. Three have a randomized design, six are non-randomized and four are observational. 

#### 3.1.5. Phases of Studies

The phases of the studies are shown in Table 6. 30% (41/137) of studies were either early Phase 1, Phase 1, or Phase 1/2. 

#### 3.1.6. Study Designs

The types of study designs are listed in Table 7. The largest proportion of studies were categorized as non-randomized (54/137 = 39%) and only 17% (23/137) were randomized with masking (blinding). 

#### 3.1.7. Types of Intervention

Table 8 lists the types of interventions used in the studies. The most frequently reported intervention was “drug” (49 interventions), followed by “biological” (38 interventions). 

#### 3.1.8. Study Participants

A histogram of the numbers of participants in the studies is shown in Figure 6. 

### 3.2. Thematic Analysis

Inspection of the titles of the studies revealed five common themes: COVID-19, treatment delivery, infection, microbiome and probiotic supplementation (Table 9).

#### 3.2.1. COVID-19

Fifteen studies included the word COVID-19 in their title (Table 10). The start dates were between 2019 and 2022, and eight countries were represented with only 53% (8/15) of the studies taking place in the USA. Most of the studies were concerned with the effects on patients and the provision of care during the disruption caused by the pandemic, but two looked at the effect of COVID-19 infection on patients with breast cancer, and one investigated the immunogenicity of a COVID-19 vaccine in patients receiving cancer treatment.

#### 3.2.2. Treatment Delivery

A total of 38 studies were concerned with vaccination and treatment using viral vectors and oncolytic viruses (Table 11). A further four studies used bacterial products (Table 12). In addition, there was one study investigating the use of a yeast-based vaccine in several cancers, including breast cancer (NCT03552718), and one study run in Spain that investigated the use of an individualized vaccination with autologous dendritic cells pulsed with the patient’s own tumor (NCT01431196).

#### 3.2.3. Infection

A total of 28 studies were concerned with infection; 18 with infection prevention and 4 with infection treatment (Table 13).

In addition, there was one study that had investigated the use of Mycograb^®^ (Efungumab, a drug developed to treat candidemia) as a treatment for advanced breast cancer (NCT00217815); a study that had investigated the natural history of fungal infections of the blood in patients with cancer (NCT00445952); a Phase 1 study studying side-effects of treatment for solid tumors in patients who also have HIV infection (NCT01249443); a study collecting material for a biorepository (NCT01931644); a study looking for an association between infection with hepatitis C virus and breast cancer (NCT04090164); and two studies investigating ways to implement breast cancer and HPV vaccination programs (NCT04638010 and NCT05524480).

#### 3.2.4. Microbiome

A total of 37 studies investigated associations between breast cancer treatments and the microbiome and body flora (Table 14).

Two of the above studies investigated the effect of breast cancer treatments on body flora. One was an observational study looking at the effects of therapies to treat tumors on changes to the flora of the blood, mouth, urethra, and intestine (NCT04202848). The other specifically investigated changes to the intestinal flora of people with breast cancer treated with a tyrosine kinase inhibitor which has a known side-effect of diarrhea (NCT05030519). 

#### 3.2.5. Probiotics

Twelve studies investigated probiotics. In ten cases the studies are single site with the PI located in the USA, and two studies took place in China.

The earliest study started in 2007 and was designed as a placebo-controlled randomized trial to assess the safety of a dietary supplement of a fermented extract of *Lactobacillus* in patients with cancer undergoing chemotherapy (NCT00606970). Unfortunately, no patients were recruited, and the study is marked as “withdrawn”.

A Phase 1 pilot study of 30 patients with a suppressed immune system due to cancer therapy used a once-daily oral dose of *Lactobacillus rhamosus* to determine if this could prevent the development of infection (NCT00946283). Unfortunately, this study was terminated early due to slow accrual.

A study run in Austria randomized 27 participants to receive either an oral probiotic supplement (capsules containing four *Lactobacillus* strains) or a placebo and aimed to improve the quality of the vaginal flora of women with breast cancer receiving treatment by chemotherapy (NCT01723592). The study concluded that the orally administered *Lactobacillus* preparation has the potential to improve the vaginal microbiota in this cohort [19].

Started in 2019, a randomized placebo-controlled trial run in Canada aims to determine whether a daily probiotic supplement containing *Lactobacillus* sp. can alter the diversity of breast microbiota in women who are at high risk of developing breast cancer (NCT03290651). A total of 60 participants were randomized by the end of 2021, but no results have been published to date. 

A study with the stated aim of investigating how probiotics will affect the subjects’ immune system during breast cancer (NCT03358511) stopped recruitment in 2020 after enrolling only seven participants in 2.5 years.

A randomized three-arm study comparing probiotic supplementation, probiotic supplementation, or matched placebo (with usual sedentary lifestyle) was designed to be run in Spain in 2018 but was withdrawn the following year with no patients enrolled (NCT03760653).

An open label, single group study of fecal microbiota transplantation as treatment for diarrhea or colitis induced by immune-checkpoint-inhibitor treatment in patients with genitourinary cancer (including breast cancer) is currently open to recruitment and aims to enroll 40 patients by 2025 (NCT04038619).

A randomized double-blind placebo-controlled trial to investigate the efficacy of a novel probiotic on the bacteriome and mycobiome of breast cancer aims to enroll 100 patients with an estimated study completion date in 2025 (NCT04362826).

A double-blind, placebo-controlled clinical trial in which breast cancer survivors experiencing moderate to severe anxiety symptoms intended to randomize to daily consumption of a synbiotic supplement (a mixture of prebiotics and probiotics) or placebo. The study terminated in 2022 with only three patients enrolled (NCT04784182).

A small pilot study investigating the effects of probiotics on the gut microbiome and immune system before surgery to remove the tumor enrolled six patients before completion in 2023 (NCT04857697).

Two studies run in China both had a start date in 2018. One study investigated the effect of probiotics on preventing patients with breast cancer from cancer-related cognitive impairment (PMID34896904), and the other was a randomized, double-blind, placebo-controlled trial that investigated probiotics for the treatment of docetaxel-related weight gain (PMID34926547). 

## 4. Discussion

This analysis of studies listed in ClinicalTrials.gov regarding the role of microorganisms in breast cancer revealed a limited number of clinical studies with a wide variety of descriptions. Only 117 studies were identified, which is less than 0.03% of the total studies listed (117/458,710). A search of PubMed yielded an additional 20 studies. Analysis of these 137 studies showed a wide geographic spread of countries where the studies took place, although most were performed in the USA and China (Table 2). Most of the studies had been completed, but almost a third were running or about to start (Table 3). However, results were only available for 10% of the studies (Table 4), a phenomenon that has been reported previously (e.g., [20]).

The earliest study start date was 1995, with an apparent increase in the number of studies started in 2018 (Figure 5). Studies started in 2022 or 2023 cover a wide range of topics, study design, and expected enrolment (Table 5).

Just under one-third (30%) of studies were in the early phase (Table 6). Only 17% of studies were randomized with masking (Table 7). Most studies had 100 or fewer participants (Figure 6).

Drawbacks of our study include the reliance on the accuracy of the data reported in ClinicalTrials.gov, as it has been shown that recruitment status (for example) can be outdated or wrong [21], and discordance has been reported with the information in the subsequent publications [22]. The studies have been supplemented with those found from a search of PubMed, but this would only include those that had been published, a potential source of publication bias [23].

The thematic analysis identified five themes: (i) COVID-19, mainly concerned with the impact of the viral COVID-19 pandemic on the care of people with breast cancer; (ii) treatment delivery, mainly concerned with vaccination and delivery of therapies for breast cancer through viral vectors; (iii) infection, studies investigating infection prevention and treatment; (iv) microbiome, studies seeking associations between breast cancer treatments and the microbiome and body flora; and (v) probiotics, studies with an intervention of probiotics (mainly oral *Lactobacillus* preparations).

Modern treatments have made breast cancer a survivable disease from which many women recover [24]; a substantial proportion return to work after some rehabilitation [25,26]. An advanced understanding of the pathology of breast cancer can potentially further improve its treatment. In addition to the microbiome on the surface of the skin, recent research has shown there to be microbes within the breast tissue [12] and that the microbial profile of the breast may be associated with the development of breast cancer [27,28].

Dysregulation of microbial homeostasis (referred to as dysbiosis) has been reported in benign and malignant breast disease [4]. The breast microbiota may be altered and influenced by microbes arising from other organs and distal sites, including urinary, oral, vaginal, and skin microbes [12,14,29,30,31]. The entero-mammary pathway has been well-established [32].

Urbaniak et al. first hypothesized the potential for a breast microbiome independent of lactation in 2012 [33] and went on to confirm the presence of breast microbiota using DNA isolation techniques [12]. Investigation into the breast microbiota as a modifiable risk factor for breast cancer has since attracted considerable scientific attention. Considering the diffuse vasculature and lymphatics, and the widespread location of the lobules and ducts leading to the nipple, it is no surprise that bacteria are widespread within the mammary gland, irrespective of lactation.

It is disappointing to see so few randomized clinical trials (RCTs), considered to provide the highest level of evidence. In the past two years, only two RCTs have been planned, one of which is an investigation of a novel probiotic on the bacteriome and mycobiome (NCT04362826, see Table 5). Probiotic supplementation is particularly well suited to being tested in RCTs, and blinding with a placebo should be a minor problem. It is hoped that funding for further work in this field will be made available to address issues such as deleterious side-effects from chemotherapy and radiation therapy on the microbiome.

However, the breadth of studies confirms the increasing scientific interest in the role of microbiota in breast cancer. We anticipate that more well-designed RCTs will lead to the development of anti-cancer therapies working with a modified microbial profile to optimize the prevention and treatment of breast cancer.

## 5. Conclusions

This analysis shows the wide variety of clinical studies concerned with microorganisms and breast cancer, covering a range of themes. Of particular interest are those studies investigating the role of microbiota as a modifiable risk factor for breast cancer, as this could lead to cost-effective methods of prevention and reduction in the severity of side-effects due to treatment.

## Figures and Tables

**Figure 1 pathogens-13-00006-f001:**
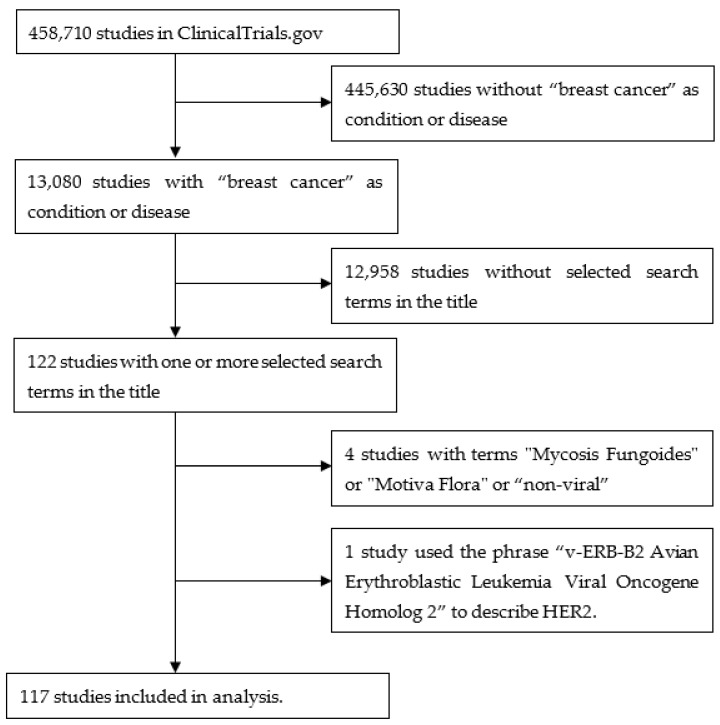
Flow diagram illustrating the steps used to obtain the studies from ClinicalTrials.gov.

**Figure 2 pathogens-13-00006-f002:**
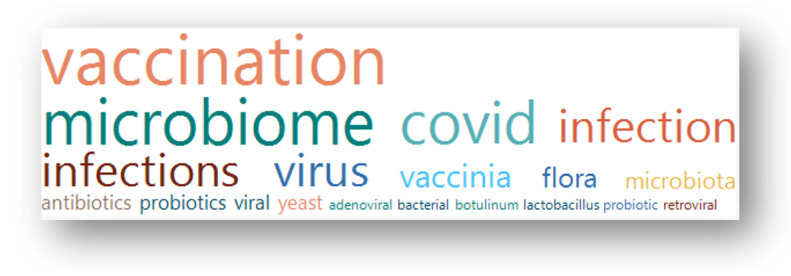
A Word Cloud visualizing the terms occurring more than once in titles of the selected studies. The size of the word is proportional to the frequency of occurrence in the term list; the colors are arbitrary.

**Figure 3 pathogens-13-00006-f003:**
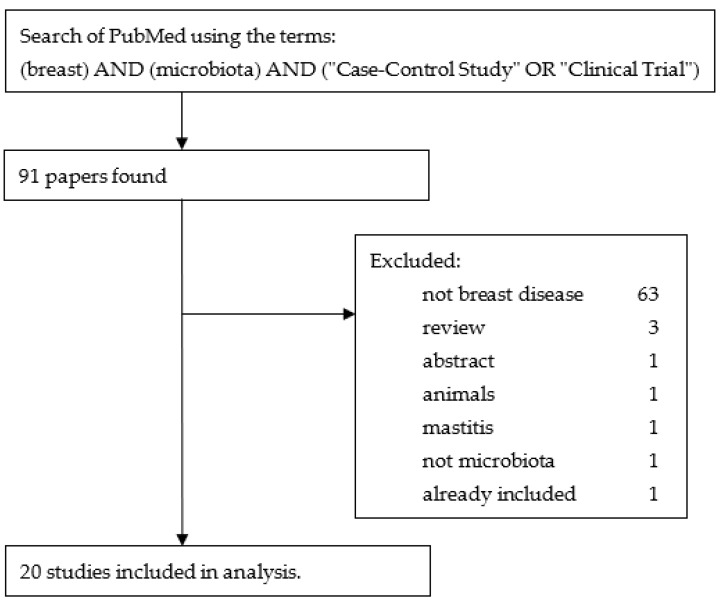
Flow diagram illustrating the steps used to obtain the studies from PubMed.

**Figure 4 pathogens-13-00006-f004:**
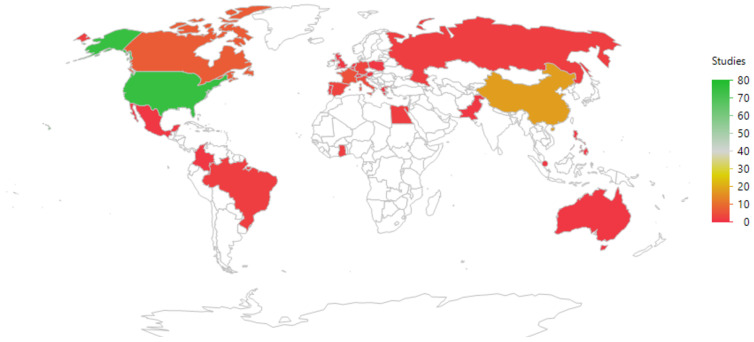
An illustration of the geographic distribution of the countries where the principal investigators were based. The color coding refers to the frequency of studies per country. The map uses the default JMP setting (a Kavrayskiy VII compromise projection).

**Figure 5 pathogens-13-00006-f005:**
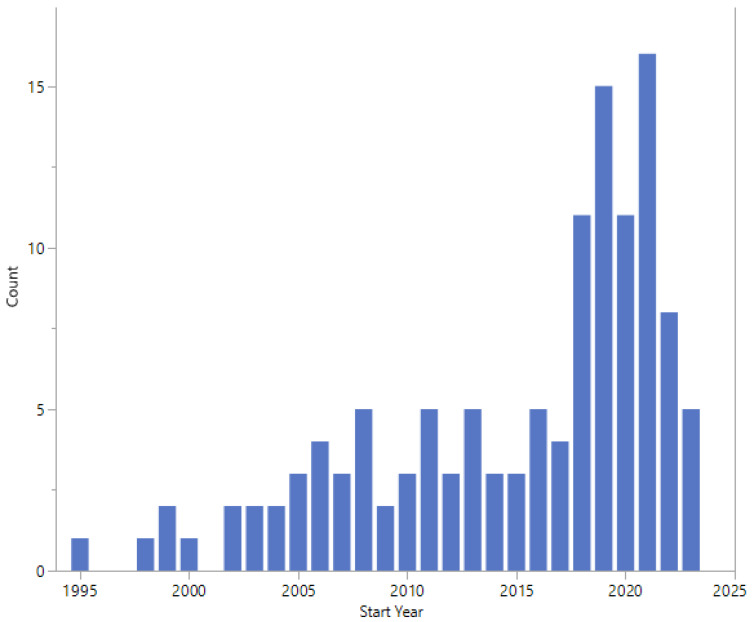
Start dates.

**Figure 6 pathogens-13-00006-f006:**
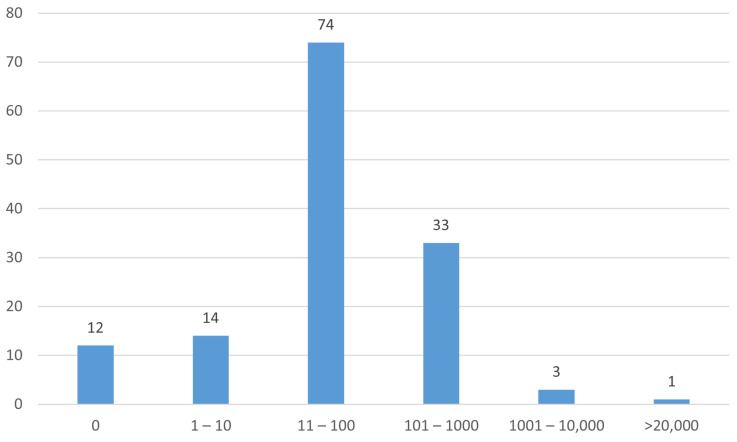
Numbers of participants (enrollment) in the studies.

**Table 1 pathogens-13-00006-t001:** A complete list of all 38 search terms and the frequency of occurrence in titles of the selected studies. Terms are sorted by Count, then alphabetically.

Term	Count
vaccination	18
microbiome	16
covid	14
infection	10
infections	10
virus	10
vaccinia	6
flora	5
microbiota	5
antibiotics	3
probiotics	3
yeast	3
adenoviral	2
bacterial	2
botulinum	2
lactobacillus	2
probiotic	2
retroviral	2
viral	2
antisepsis	1
antiseptics	1
aspergillosis	1
biotics	1
clostridium	1
coxsackie	1
fecal	1
infected	1
infectious	1
influenza	1
microbe	1
microbiotaã ^1^	1
mycobiome	1
mycograb	1
mycosis	1
pneumococcus	1
pneumonia	1
poxviral	1
saccharomyces	1

^1^ This term is the result of an unusual character combination that has been interpreted as a special character.

**Table 2 pathogens-13-00006-t002:** A complete list of countries where the principal investigator was based for all 137 studies. The countries were sorted by number of studies, then alphabetically.

Country	Studies
USA	75
China	18
Canada	7
France	5
Italy	4
Germany	3
Spain	3
Brazil	2
Egypt	2
Ghana	2
Russia	2
Australia	1
Austria	1
Belgium	1
Colombia	1
Greece	1
Mexico	1
Netherlands	1
Pakistan	1
Philippines	1
Poland	1
Portugal	1
Singapore	1
Switzerland	1
United Kingdom	1

**Table 3 pathogens-13-00006-t003:** Study Status.

Study Status	Number of Studies
Not yet recruiting	3
Recruiting	31
Active not recruiting	10
Completed	70
Approved for marketing	1
Terminated	8
Unknown	8
Withdrawn	6
Total	137

**Table 4 pathogens-13-00006-t004:** Studies with results available.

Results Available?	Number of Studies
No	108
Yes	29
Total	137

**Table 5 pathogens-13-00006-t005:** Studies with a planned start date in 2022 or 2023 (grouped by study design, sorted by status).

Study Design and Topic	Status *	Planned Enrollment	NCT Number
Randomized			
Novel Probiotics and the bacteriome and mycobiome	NYR	100	NCT04362826
Diagnostic evaluation during the COVID-19 pandemic	Recruiting	196	NCT05181722
Randomized (cluster)			
Cancer screening and HPV vaccination	NYR	2000	NCT05524480
Non-randomized			
Intratumoral oncolytic virus	NYR	24	NCT05600582
Oncolytic virus injection	Recruiting	20	NCT05860374
Recombinant herpes simplex virus I	Recruiting	24	NCT05886075
MEM-288 oncolytic virus	Recruiting	18	NCT05076760
Vaccinia virus VV-GMCSF-Lact	Recruiting	73	NCT05376527
WOKVAC vaccine	Recruiting	16	NCT04329065
Observational			
Oral Aromatase Inhibitors and Gut Microbiome	Recruiting	25	NCT05030038
COVID-19 related financial hardship	ANR	14	NCT05076266
Gut microbiome components	Recruiting	100	NCT05444647
Intestinal microbiota	Recruiting	35	NCT05580887

* ANR: Active Not Recruiting; NYR: Not Yet Recruiting.

**Table 6 pathogens-13-00006-t006:** Phase of study.

Phase of Study	Number of Studies
Early Phase 1	5
Phase 1	28
Phase 1/Phase 2	8
Phase 2	13
Phase 3	7
Phase 4	5
NA ^1^	22
Not specified ^2^	49
Total	137

^1^ Phase Not Applicable, such as trials of devices or behavioral interventions. ^2^ Phase was not specified.

**Table 7 pathogens-13-00006-t007:** Design of study.

Design of Study	Number of Studies
Randomized with masking	23
Randomized without masking	17
Non-randomized	54
Observational	42
Not specified	1
Total	137

**Table 8 pathogens-13-00006-t008:** Interventions used in the studies.

Intervention	Number of Studies
Behavioral	3
Biological	38
Device	2
Diagnostic test	1
Dietary supplement	10
Drug	49
Genetic	4
Other	21
Procedure	8
Radiation	3
Not applicable (observational study)	22

NOTE: Some studies have more than one intervention (e.g., biological plus drug).

**Table 9 pathogens-13-00006-t009:** Themes arising from the analysis.

Theme	Number of Studies
COVID-19	15
Treatment delivery	44
Infection	29
Microbiome	37
Probiotic supplements	12
	137

**Table 10 pathogens-13-00006-t010:** Studies that included COVID-19 in the title (sorted by start date).

Study Description	Location	Start Date	Identifier *
Impact of the pandemic on patient economic factors	USA	2019	NCT04169542
Impact of COVID-19 infection in women with cancer	France	2020	NCT04351139
Cancer screening and prevention during the pandemic	USA	2020	NCT04587258
Nutritional care in oncology patients during the pandemic	Greece	2020	NCT04876560
Effect of the pandemic on management of patients with breast cancer	Pakistan	2020	NCT04929964
Effect of one preoperative fraction of radiation during the pandemic	Canada	2020	NCT05037019
A survey of cancer patient perspectives during the pandemic	USA	2020	NCT05062538
Remote rehabilitation in women with breast cancer during the pandemic	Brazil	2020	NCT05530876
Changes in gut microbiota composition after 12 weeks in lockdown	Italy	2020	PMID37727203
Immunogenicity of COVID-19 vaccine in patients receiving cancer treatment	USA	2021	NCT04821570
Evaluating treatment for COVID-19 infection in breast cancer patients	Egypt	2021	NCT04871854
Patient experiences with COVID-19 vaccination after breast cancer treatment	USA	2021	NCT04872738
Impact of the pandemic on patient economic factors	USA	2022	NCT05076266
Timely diagnostic evaluation during the pandemic	USA	2022	NCT05181722
At-home administration of chemotherapy during the COVID-19 pandemic	USA	NS	NCT04395508

* prefix NCT is from ClinicalTrials.gov, prefix PMID is from PubMed; NS: start date not specified.

**Table 11 pathogens-13-00006-t011:** Studies investigating vaccination, and treatment with viral vectors and oncolytic viruses (grouped by investigation type then sorted by NCT Number).

Investigation Type	Number of Studies	NCT Number
Oncolytic virus	15	NCT00574977, NCT00636558, NCT01152398, NCT01846091, NCT02179515, NCT03004183, NCT03110445, NCT03740256, NCT04215146, NCT05076760, NCT05180851, NCT05376527, NCT05600582, NCT05860374, NCT05886075
Vaccination	17	NCT00027131, NCT00197522, NCT00317603, NCT00485277, NCT00622401, NCT00880464, NCT00924092, NCT01127074, NCT01291420, NCT02276300, NCT02938442, NCT03632941, NCT03789097, NCT04105582, NCT04329065, NCT01390064, NCT02395614
Viral vector gene transfer	6	NCT00001493, NCT00307229, NCT00451022, NCT01703754, NCT02140996, NCT02576665

**Table 12 pathogens-13-00006-t012:** Studies investigating treatment with bacterial products (sorted by start date).

Study Description	PI Location	Start Date	NCT Number
Use of botulinum toxin A in breast reconstruction	Canada	2011	NCT01427400
Botulinum toxin A in tissue expander breast reconstruction	USA	2012	NCT01591746
Pembrolizumab with intratumoral injection of *Clostridium novyi*-NT	USA	2018	NCT03435952
Bacterial cellulose-monolaurin hydrogel for acute radiation dermatitis	Philippines	2021	NCT05079763

**Table 13 pathogens-13-00006-t013:** Studies investigating infection prevention and treatment (grouped by investigation type then sorted by NCT Number).

Investigation Type	Number of Studies	Identifier *
Infection Prevention	18	NCT00003883, NCT00005590, NCT00045292, NCT00064311, NCT00079222, NCT00324324, NCT00378781, NCT00536081, NCT00741039, NCT01286168, NCT01899690, NCT02395614, NCT02479347, NCT02816112, NCT03229824, NCT03742908, NCT04818931, PMID37754546
Infection Treatment	4	NCT00014391, NCT00110045, NCT00509691, NCT00769613

* prefix NCT is from ClinicalTrials.gov, prefix PMID is from PubMed.

**Table 14 pathogens-13-00006-t014:** Studies investigating associations between treatment for breast cancer and the microbiome and body flora (sorted by start date).

Study Description	PI Location	Start Date	Identifier *
Intratumoral microbiome is driven by metastatic site	France	2012	PMID36868056
Analysis of gut microbiome predicts risk of diarrhea associated with neratinib	USA	2015	PMID33796451
Gut microbiome and gastrointestinal toxicities after neoadjuvant chemotherapy	USA	2016	NCT02696759
Effect of radiotherapy on circulating immune cells and effect on microbiome	USA	2018	NCT03383107
The role of the skin microbiome in post-mastectomy radiation dermatitis	USA	2018	NCT03519438
Gut and intratumoral microbiome effect on neoadjuvant chemotherapy	USA	2017	NCT03586297
Relationship between gut microbiome and adjuvant chemotherapy	China	2018	NCT03702868
Breast cancer and its relationship with the microbiota	Spain	2018	NCT03885648
Effects of exercise on gut microbe composition in breast cancer survivors	USA	2020	NCT04088708
Study of skin microbiome after chemotherapy for breast cancer	China	2019	NCT04132713
Intestinal microbiota of patients with breast cancer undergoing chemotherapy	China	2019	NCT04138979
A study of modern therapies on flora in body fluids and blood	China	2019	NCT04202848
Assessing the impact of the microbiome on breast cancer radiotherapy toxicity	USA	2019	NCT04245150
Microbiota as a non-invasive tool to predict postoperative depression	China	2019	PMID33211236
Breast microbiome associations with breast tumor characteristics	China	2019	PMID36172155
The role of gut microbiota in women treated with aromatase inhibitors	Italy	2019	PMID36558756
Preoperative gut microbiota and chronic postoperative pain	China	2019	PMID34403381
Effect of an anaesthetic given during breast cancer surgery on gut microbiota	China	2021	NCT04303325
Exercise, gut microbiome, and breast cancer in underserved populations	USA	2021	NCT05000502
Microbiome and association with breast implant infections	USA	2021	NCT05020574
Oral aromatase inhibitors and the gut microbiome	USA	2022	NCT05030038
A study of diarrhea and intestinal flora changes after a breast cancer therapy	China	2021	NCT05030519
The association between radiation dermatitis and skin microbiome	China	2021	NCT05032768
The gut microbiome and immune-checkpoint-inhibitor therapy	USA	2021	NCT05037825
Changes in the gut microbiome and chemotherapy-induced nausea	USA	2021	NCT05417867
Mechanism of acupuncture on cancer-related fatigue	China	2021	PMID35578688
A study of gut microbiome components and response to neoadjuvant therapy	China	2022	NCT05444647
Intestinal microbiota impact on prognosis and treatment outcomes	Russia	2022	NCT05580887
Breast cancer survivors and healthy women: “BiotaCancerSurvivors” study	Portugal	NS	PMID36765550
Fecal microbiota composition in patients with breast cancer	France	NS	PMID34444865
The oral microbiome and breast cancer in the Ghana Breast Health Study	Ghana	NS	PMID35657343
Fecal microbial profiles and breast cancer in the Ghana Breast Health Study	Ghana	NS	PMID33460452
Diet alters entero-mammary signaling to regulate the breast microbiome	USA	NS	PMID34083249
Plasma metabolomic signatures associated with long-term breast cancer risk	France	NS	PMID31164347
Diet-related metabolomic signature of long-term breast cancer risk	France	NS	PMID31767565
Health-related quality of life is associated with fecal microbial composition	USA	NS	PMID36512109
Potential antiproliferative activity of polyphenol metabolites	Brazil	NS	PMID28541359

* prefix NCT is from ClinicalTrials.gov, prefix PMID is from PubMed; NS: start date not specified.

## Data Availability

All data used in this work can be found in Appendix A.

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
