# Peer review of "Microorganisms and Breast Cancer: An In-Depth Analysis of Clinical Studies"

_pathogens, 2023, doi:10.3390/pathogens13010006_

Round 1

Reviewer 1 Report

Comments and Suggestions for Authors

Naderi et al. primary focus was to scrutinize clinical studies listed on ClinicalTrials.gov. They utilized specific search terms associated with bacteria, viruses, and fungi. These studies were categorized into thematic groups to identify knowledge gaps and emerging trends. This research approach was systematic and thorough, with the aim of shedding light on the connection between microorganisms and breast cancer.

major concerns:

·      The article lacks original data.

·      In its current form, the article is more suitable for submission as a review rather than being accepted for publication.

Comments on the Quality of English Language

Minor editing of English language required

Author Response

Reviewer 1

Naderi et al. primary focus was to scrutinize clinical studies listed on ClinicalTrials.gov. They utilized specific search terms associated with bacteria, viruses, and fungi. These studies were categorized into thematic groups to identify knowledge gaps and emerging trends. This research approach was systematic and thorough, with the aim of shedding light on the connection between microorganisms and breast cancer.

major concerns:

  • The article lacks original data.

The authors thank the reviewer for allowing us to make the point that while our manuscript can be viewed as simply a summary of published data, we feel that there is much original work, namely:

  • An initial search of ClinicalTrials.gov yielded only 25 studies, but in the authors’ experience this was clearly far too few. We therefore created a bespoke search phraseology from all 13,080 studies of “breast cancer” on the database, using 10,729 terms. We were able to identify 38 terms that when applied in a search increased the number of studies identified from 25 to 117. All the search terms are listed in the article and can be used in future by other researchers if they wish.
  • Further to a reviewer’s request, we have now included the results from a PubMed search. The 71 papers identified were reduced to 20 after detailed consideration of each. Therefore, we have found, from two different sources, a total of (117+20=) 137 studies for inclusion in our article.
  • Characteristics of these studies that have been analysed include geographic distribution, recruitment status, the distribution of start dates, phase/design/interventions used, and numbers of subjects enrolled.
  • Detailed examination of the 137 studies revealed that they can be grouped into five “themes” (Table 9).

In summary, we think the search methodology and analyses of the studies identified are unique and original and will be of interest to readers of this journal.

  • In its current form, the article is more suitable for submission as a review rather than being accepted for publication.

Please see the comments made in the preceding section. We think the generation of a bespoke set of search terms, and analyses of related studies identified, should be viewed as original work that will fit into the remit of this Special Issue and will complement the other articles.

We have included a revised draft manuscript incorporating the changes from all reviewers (highlighted in yellow) so that the responses can be seen in context.

Reviewer 2 Report

Comments and Suggestions for Authors

The authors were interested in collecting clinical studies dissecting the interactions of macroorganisms and breast cancer. They have put coonsiderable effort in the work, but it is unlikely that potential users will also be interested in all aspects of this interacion; the scope of their research will probably be much narrower, therefore they can easily make an analogous, but narrower literature search on their own. In conlusion, the effort of authors is respectable, but the inerest of the review is rather low.

1. What is the main question addressed by the research?
The authors were interested in collecting clinical studies dissecting the interactions of microorganisms and breast cancer.

2. Do you consider the topic original or relevant in the field? Does it
address a specific gap in the field?
It is a new idea to collect all clinical studies that, in some form, investigate the interaction of microorganisms and breast cancer. It is unlikely, however, that the potential readers will share this broad interest, they would rather concentrate on specific field.

3. What does it add to the subject area compared with other published
material?
It reviews together the different questions and approaches of oncology and microbes. Their collection is basically correct, but a more detailed theoretical introduction might improve the general interest of the paper.

4. What specific improvements should the authors consider regarding the
methodology? What further controls should be considered?
The methodology of the review is correct.

5. Are the conclusions consistent with the evidence and arguments presented
and do they address the main question posed?
The authors did not reach a specific conclusion, they rather "list" the ongoing research efforts of others.

6. Are the references appropriate?
Yes.

7. Please include any additional comments on the tables and figures.
The tables are correct.     

Author Response

Reviewer 2

The authors were interested in collecting clinical studies dissecting the interactions of macroorganisms and breast cancer. They have put coonsiderable effort in the work, but it is unlikely that potential users will also be interested in all aspects of this interacion; the scope of their research will probably be much narrower, therefore they can easily make an analogous, but narrower literature search on their own. In conlusion, the effort of authors is respectable, but the inerest of the review is rather low.

  1. What is the main question addressed by the research?

The authors were interested in collecting clinical studies dissecting the interactions of microorganisms and breast cancer.

  1. Do you consider the topic original or relevant in the field? Does it address a specific gap in the field?

It is a new idea to collect all clinical studies that, in some form, investigate the interaction of microorganisms and breast cancer. It is unlikely, however, that the potential readers will share this broad interest, they would rather concentrate on specific field.

Thank you for giving us the opportunity to explain our rationale. We think that readers of the Special Issue entitled “Role of Microorganisms in Breast Cancer" will appreciate an article summarising the clinical studies, both published and unpublished, and will enable them to view the “state of the art” of their specific field in the context of similar work.

  1. What does it add to the subject area compared with other published material?

It reviews together the different questions and approaches of oncology and microbes. Their collection is basically correct, but a more detailed theoretical introduction might improve the general interest of the paper.

The introduction has been modified to make it more general.

  1. What specific improvements should the authors consider regarding the methodology? What further controls should be considered?

The methodology of the review is correct.

Thank you.

  1. Are the conclusions consistent with the evidence and arguments presented and do they address the main question posed?

The authors did not reach a specific conclusion, they rather "list" the ongoing research efforts of others.

Some conclusions have been added.

  1. Are the references appropriate?

Yes.

Thank you.

  1. Please include any additional comments on the tables and figures.

The tables are correct.

Thank you.

We have included a revised draft manuscript incorporating the changes from all reviewers (highlighted in yellow) so that the responses can be seen in context.

Reviewer 3 Report

Comments and Suggestions for Authors

Dear reviewers,

Congrats on the extensive review regarding data on microbiota and breast cancer. However, some issues must be addressed/ameliorated before the article can be considered for final publication:

1) English review

2) You need to perform, as well, a literature review on data studies published that are outside the clinicaltrials.gov database. For that, I suggest specific "mesh" words" and search engines, like PubMed or Medline, CINAHAL, Cochrane, and even Google Academic. An example of a manuscript published in "Cancers" MDPI: "Caleça T, et al. Breast Cancer Survivors and Healthy Women: Could Gut Microbiota Make a Difference?-"BiotaCancerSurvivors": A Case-Control Study. Cancers (Basel). 2023 Jan 18;15(3):594. doi: 10.3390/cancers15030594. PMID: 36765550; PMCID: PMC9913170."

3) We need a table resume of the relevant results on microbiota and breast cancer studies

4) A better narrative review to improve the introduction and discussion contents.

Kind regards,

Comments on the Quality of English Language

English minor review

Author Response

Reviewer 3

Congrats on the extensive review regarding data on microbiota and breast cancer. However, some issues must be addressed/ameliorated before the article can be considered for final publication:

1) English review

The authors have checked the English line-by-line and made changes as required.

2) You need to perform, as well, a literature review on data studies published that are outside the clinicaltrials.gov database. For that, I suggest specific "mesh" words" and search engines, like PubMed or Medline, CINAHAL, Cochrane, and even Google Academic. An example of a manuscript published in "Cancers" MDPI: "Caleça T, et al. Breast Cancer Survivors and Healthy Women: Could Gut Microbiota Make a Difference?-"BiotaCancerSurvivors": A Case-Control Study. Cancers (Basel). 2023 Jan 18;15(3):594. doi: 10.3390/cancers15030594. PMID: 36765550; PMCID: PMC9913170."

Thank you for this suggestion. A literature review using the terms (breast) AND (microbiota) AND ("Case-Control Study" OR "Clinical Trial") in PubMed yielded 91 results. Of these, 71 were excluded, leaving 20 to be added (including the paper indicated by the referee). The text, tables, figures, and supplementary file have been amended accordingly.

3) We need a table resume of the relevant results on microbiota and breast cancer studies

A Table has been created to summarise the themes found from the analysis of the studies (Table 9 line 221). Of the 37 studies investigating the microbiome, only 15 have reported results. As this is a possible source of publication bias, we are reluctant report these results. We understand that this article (if accepted) will be part of a Special Issue entitled “Role of Microorganisms in Breast Cancer", so presumably the other papers will discuss results in detail.

4) A better narrative review to improve the introduction and discussion contents.

The introduction and discussion have been modified.

We have included a revised draft manuscript incorporating the changes from all reviewers (highlighted in yellow) so that the responses can be seen in context.

Round 2

Reviewer 2 Report

Comments and Suggestions for Authors

Since the authors conducted additional research and added further backgroud infomation, the manuscript considerably improved. Now it provides an interesting overview of the directions of current investigations  on breast cancer and microbiobes, including cutting adge reserch on the role of microbiome.